# A Comprehensive Analysis of Unclamped-Inductive-Switching-Induced Electrical Parameter Degradations and Optimizations for 4H-SiC Trench Metal-Oxide-Semiconductor Field-Effect Transistor Structures

**DOI:** 10.3390/mi15060772

**Published:** 2024-06-09

**Authors:** Li Liu, Jingqi Guo, Yiheng Shi, Kai Zeng, Gangpeng Li

**Affiliations:** 1State Key Laboratory of Wide-Bandgap Semiconductor Devices and Integrated Technology, School of Microelectronics, Xidian University, Xi’an 710071, China; 22111213670@stu.xidian.edu.cn; 2Guangzhou Institute of Technology, Xidian University, Guangzhou 510555, China

**Keywords:** SiC trench power MOSFET, UIS, failure analysis, optimized structure

## Abstract

This paper presents a comprehensive study on single- and repetitive-frequency UIS characteristics of 1200 V asymmetric (AT) and double trench silicon carbide (DT-SiC) metal-oxide-semiconductor field-effect transistors (MOSFETs) and their electrical degradation under electrical–thermal working conditions, investigated through experiment and simulation verification. Because their structure is different, the failure mechanisms are different. Comparatively, the gate oxide of a DT-MOSFET is more easily damaged than an AT-MOSFET because the hot carriers are injected into the oxide. The parameters’ degradation under repetitive UIS stress also requires analysis. The variations in the measured parameters are recorded to evaluate typical electrical features of device failure. Furthermore, TCAD simulation is used to reveal the electrothermal stress inside the device during avalanche. Additionally, failed devices are decapsulated to verify the location of the failure point. Finally, a new type of stepped-oxide vertical power DT MOSFET with P-type shielding and current spread layers, along with its feasible process flow, is proposed for the improvement of gate dielectric reliability.

## 1. Introduction

Due to its wide bandgap, highly critical breakdown electrical field, and thermal conductivity, SiC is recognized as the most promising candidate for high-temperature and high-power applications [1,2,3].

Because intrinsic SiO_2_ is used as the gate dielectric, SiC vertical power MOSFETs possess simultaneous advantages of high switching speed and high power, enabling the devices to be widely applied in power market areas such as photovoltaic inverters, electric vehicles, electric aircraft, uninterruptible power supplies, and energy distribution networks. Under these application scenarios, high dv/dt and di/dt [4] could be generated during high-speed switching. If the inductive load is used, an induced voltage will be generated and stressed between the drain and source of the DUT during the fast turn-on and turn-off transient because the variation in the current that flows through the load inductor cannot track the one in loop synchronously. If the induced voltage exceeds the breakdown voltage of the devices, the devices will be forced into avalanche mode. The energy stored in the load inductor will be released through the power devices at the turn-off point. Simultaneous high voltage and large current can stress the devices, potentially leading to the failure of the devices [5,6,7].

Until now, many researchers have focused on UIS phenomena of planar MOSFETs. It is generally believed that there are two possible reasons for a single UIS failure of the device [8,9,10,11,12,13,14,15,16,17,18,19]: One is the turn-on of parasitic BJT triggering positive thermal-electrical feedback, ultimately leading to device failure. The other is high junction temperature causing the aluminum metal to melt, resulting in device failure. As to a repetitive UIS failure mechanism, it is commonly believed that a hot hole in the JFET region will be injected into oxide and trapped by the interface states and gate oxide leading to the parameters’ degradation. However, there are fewer comprehensive studies [7,20,21] on UIS failure mechanisms including temperature characteristics and some important circuit loop parameters under single and repetitive UIS stress on trench gate MOSFETs.

In this paper, a comprehensive study of single- and repetitive-frequency UIS characteristics of 1200 V AT-MOSFETs and DT-SiC MOSFETs under different working conditions are investigated by experiment and simulation, i.e., Rohm DT-MOSFETs and Infineon AT-MOSFETs with the same current rate and on-state resistance. The single-pulse UIS and repetitive UIS measurements were performed on these two kinds of devices. Degradation and failure mechanisms of devices have been researched through the combination of measured data, numerical simulation, and microstructure analysis. A reasonable physical failure explanation during avalanche has been given. Additionally, failed devices have been decapsulated to verify the location of the failure point from the perspective of the semiconductor die. Finally, a new type of vertical power DT MOSFET is proposed for the improvement of the gate’s dielectric reliability.

## 2. Device Structures and Experimental Setup

The measurements were taken using devices under test (DUTs) listed as Rohm SCT3080 as shown in Figure 1a as a symmetrical double-trench SiC MOSFET rated at 4.2 V, 30 A, 1200 V, and 80 mΩ with an area of 2.4 mm × 3.0 mm. IMZ120R090M1HXKSA1 is shown in Figure 1b as an asymmetrical trench SiC MOSFET rated at 4.5 V, 26 A, 1200 V, and 90 mΩ with an area of 2.2 mm × 2.1 mm. All were encapsulated in TO-247-3 packages.

Figure 2 shows the circuit diagram (a) and wave curve diagrams (b). Figure 2c shows the image of UIS test setup. The main circuit includes two parts: driver circuit and power loop. The driver circuit is composed of driver source, gate resistance RG, and the DUT. The power loop includes a DC voltage source VDD, a DC-stabilizing capacitor CDC, a load inductor L, and the DUT. Both DUT and Si IGBT act as charging switch to control load inductor L. Si IGBT will isolate DUT from DC bus. In this circuit, when IGBT module and VG turn off simultaneously, it ensures that DC bus does not participate in energy dissipating during avalanche events. During the whole measurement, VDD could be set much higher than the avalanche voltage of DUT; thus, the conduction time can be decreased and thereby helps to reduce heat generation.

Figure 2b depicts the waveform diagrams where BVDSS is avalanche voltage, t_3_–t_2_ is avalanche time tav, and t_2_–t_1_ is charging time of load inductor. The DC power supply voltage was set to 50 V. The gate voltage was switched from −3 to +15 V, several 20 Ω resistors were parallelly connected between the gate driver and MOSFET, and the load inductor was chosen to be 3 mH. As can be seen from Figure 2a and the waveforms in Figure 2b, when the IGBT and DUT turn on at the same moment, the current flowing through the inductor increases linearly. When the pre-set maximum current value is achieved, the energy stored in inductor forces DUT to enter avalanche mode simultaneously when the DUT turns off.

## 3. Parameter Degradation and Avalanche Ruggedness Analysis

### 3.1. Single UIS Test

Figure 3 and Figure 4 show the typical single UIS waveforms of AT-MOSFETs and DT-MOSFETs during the last test before failure (Figure 3) and at failure (Figure 4) under 3.0 mH of inductive load conditions, respectively. When the MOSFET turns on, the I_DS_ rises linearly to the maximum drain-source current (I_DS, Max_) with a slope of V_DD_/L. After it turns off, the energy stored in the inductive load forces the MOSFET to enter into avalanche mode. At this moment, the V_DS_ maintains breakdown voltage and the I_DS_ gradually decreases to zero from I_DS, max._ For AT-MOSFETs, the peak drain current and the voltage across the drain to the source are 18 A and 1820 V during avalanche, respectively. The avalanche state lasts for 32.48 us, while the gate pulse width reaches 1.08 ms. Avalanche energy is 0.53 J. When avalanche occurs, as shown in Figure 4b, the gate pulse width increase to 1.14 ms, the peak drain current increase to 19 A, and avalanche voltage remains 1820 V. The avalanche time lasts for 18.45 us, then the DUT fails, which means that the blocking capability is lost between the source and the drain. Comparatively, for DT-MOSFETs, before failure, the avalanche current and voltage are respectively 17 A and 2280 V, the avalanche state lasts for about 24.29 us, and the gate pulse width reaches 1.02 ms. Avalanche energy is 0.47 J. When avalanche occurs, the current reaches 18 A and the gate pulse width reaches 1.08 ms, while the avalanche state lasts for only 30.45 ns because of the longer gate pulse width than the one before failure. At this moment, more avalanche energy is injected into the parasitic pn-diode, and the pn-diode cannot withstand it. Then, after the avalanche state has lasted for only 30.45 ns, the device fails. In Figure 4a,b, because of the IGBT module protection (as marked with a black arrow), thermal away did not occur in our test and the current increase was shut down by the turn-off of the IGBT module.

Table 1 presents the I_DS,Max_, V_DS,max_, E_AV,_ and t_AV_ of the DUTs under the single UIS test with different inductive loads. We can see from this table that the avalanche withstanding capability gradually increases with the increase in inductor load. The peak avalanche current presents a decreasing trend, while the avalanche duration gradually increases.

The E_AV_ of AT-MOSFETs is higher than that of DT-MOSFETs, which means superior single UIS ruggedness than SiC MOSFETs with an AT structure. Also, higher E_AV_ indicates serious thermal stress in SiC MOSFETs, so the avalanche voltage of 2240 V makes DT-MOSFETs more susceptible than AT-MOSFETs to gate oxide reliability.

Figure 5 gives the dependence of t_AV_ and E_AV_ on on-state time t_on_ for both AT-MOSFETs (a) and DT-MOSFETs (b) with a fixed load inductor value. In these two figures, a symbol plus a dashed line represents t_AV_ versus t_on_ and a symbol plus a solid line represents E_av_ versus t_on_. Different temperature characteristics can be distinguished by a black, solid square (25 °C); a red, solid circle (75 °C); a blue, solid, right-way-up triangle (125 °C); and a green, solid, upside-down triangle (175 °C). All measurement dates are recorded before the DUT failure occurs. In our measurements, since DT- and AT-MOSFETs have the same electrical characteristics ratio regarding breakdown voltage and current, the maximum current I_DS,Max_ flowing inductor when avalanche occurs also has the same value. We can see from Figure 2b that t_AV_ ∝ 1/BV, and BV increases with increasing temperature, and thus t_AV_ will decrease a little with the increasing temperature, linearly proportional to t_on_. E_AV_ is exponential to t_on_, which is pre-set by the tester and is independent of temperature, so E_AV_ overlaps for both devices in the temperature range.

Figure 6 presents the dependence of t_av_ and E_av_ for DT- and AT-SiC MOSFETs on ambient temperature with a load inductor equal to 3 mH in UIS conditions. For these two types of transistors, a different variation with temperature is observed. For AT-MOSFETs, t_av_ and E_av_ decrease a little bit as the temperature increases. From 25 °C to 175 °C, the maximum t_av_ decreased by approximately 3% and the maximum E_av_ decreased by about 6%. While, for DT-MOSFETs, comparatively, as the ambient temperature increases, both t_av_ and E_av_ increase monotonically. From 25 °C to 175 °C, t_av_ increased by approximately 7% and the maximum E_av_ increased significantly, by 26%. So, the temperature has a significant impact on the avalanche energy of DT MOSFETs.

Figure 7 shows the peak avalanche current (I_peak_), start point of avalanche voltage (BVst), and peak avalanche voltage (V_peak_) at failure of DT MOSFETs and AT MOSFETs with a fixed 3 mH load inductor. This figure shows that, for AT MOSFETs, the peak point of avalanche voltage and start point of avalanche voltage has increased as the ambient temperature increased, while the peak avalanche current at failure has decreased first and then stayed almost constant. For DT MOSFETs, these three electrical parameters all show a monotonously increasing trend with temperature. The start point and peak point avalanche voltage also increase monotonously.

Figure 8 shows the decapsulation images and failure point of AT-MOSFETs and DT-MOSFETs under a single UIS test. For AT-MOSFETs, the failure point is obviously located at the point beside the gate electrode and on the source electrode.

Figure 8b,c present the decapsulation images and failure points of a DT-MOSFET at 25 °C and 175 °C. During our measurements, when the gate driver signal is switched off, the gate voltage does not immediately drop to zero, which is marked in Figure 4a, and this phenomenon indicates that the avalanche current is flowing not only from drain to source, but also through the gate to the source, leading to abnormal gate voltage. Thus, the DT-MOSFET is susceptible to gate oxide damage. As demonstrated in Figure 8a,b, an obvious crack is found above the gate electrode. It can be thought that the failure is because of gate oxide damage [10,22,23].

### 3.2. Repetitive Avalanche Stress

Figure 9 shows the waveforms of a DT-MOSFET and an AT-MOSFET under repetitive avalanche stress. The avalanche energy of each cycle under a repetitive UIS test is set to 20%, with 40% Eav for single UIS. In our experiment, since the experiment phenomena and degradation failure mechanisms are similar to that of a 20% energy ratio, the waveforms and parameter degradations under a 40% energy ratio are not shown here. V_cont_ and V_G_ are, respectively, the control signals of an IGBT module and gate pulse signal. I_D_ and BV_DSS_ are the inductor current and avalanche voltage, respectively. During the repetitive pulse experiment, a heat sink and fan are used to help with thermal dissipating. The width of the pulse is set to 200 ms under a 20% energy ratio, while the pulse width is 500 ms under a 40% energy ratio. Before our experiment, a pretest confirmed that the pulse width and the heating sink settlement can effectively dissipate the heat from the device.

Figure 10 and Figure 11 give the threshold voltage V_th_, on-state resistance R_on_, gate leakage current I_gss,_ and forward-voltage of body diode V_SD_ at room temperature versus UIS cycles under a 20% avalanche energy ratio with a load inductance of 3 mH for the AT-MOSFET and for the DT-MOSFET. V_th_ is tested under V_DS_ = 10 V and I_D_ = 5 mA, while R_on_ is tested under V_GS_ = 18 V and I_D_ = 10 A. Gate leakage I_gss_ is tested with V_GS_ = 22 V and V_DS_ = 0 V, and V_sd_ is tested with V_GS_ = 0 V and I_SD_ = 8.5 A. Combined with Figure 10a,b, for the DT-MOSFET, V_th_ and R_on_ sharply decrease to 3.9 V and 73 mΩ·cm^2^, respectively, at the beginning, and then as the cycles increase, the values remain almost unchanged. While, for the AT-MOSFET, these two electrical parameters almost keep constant after repetitive UIS pulses, which indicates that the gate oxide has been well protected by the P+ region in the source of the AT-MOSFET. The strong electrical field under the corner of the gate trench oxide is transferred to the P+ shielding region, thus avoiding the gate oxide dielectric breakdown in advance.

Figure 11a,b present the gate leakage current (I_gss_) and forward voltage of the body diode (V_sd_) of the AT-MOSFET and the DT-MOSFET under a 20% avalanche energy ratio. In Figure 11a,b, for the DT-MOSFET, the I_gss_ increased suddenly after 10 K cycles while the V_SD_ dropped at the beginning. The avalanche has affected both the gate oxide and the body diode of the device. For the AT-MOSFET, the I_gss_ almost remained constant after repetitive UIS pulses, which has been reflected in Figure 10a,b where the V_th_ and R_on_ almost remained constant during the cycling test. While in Figure 11b, we can see that UIS cycles had much less of an effect on the V_SD_ of the body diode.

### 3.3. TCAD Simulation under Repetitive UIS Stress

Figure 12 gives the electrical field profile and impaction ionization (I.I) of the AT-MOSFET and the DT-MOSFET under a 20% avalanche energy ratio with a load inductor of 3 mH. The gate conduction time was set to 500 us and 400 us, respectively. Sentaurus TCAD mixed-mode simulation with coupled electro-thermal conditions was used to investigate the avalanche failure mechanism under repetitive UIS stress. For the AT-MOSFET, the thickness and doping concentration of the drift region were set at 10 μm and 1.2 × 10^15^ cm^−3^, respectively. The thickness of the gate oxide at the trench sidewalls and trench bottom was set at 60 nm and 100 nm, respectively. The channel length was 0.5 um with P-type doping 1 × 10^17^ cm^−3^. For the DT-MOSFET, the gate trench and source trench were the same size with a width and depth of 1 um. The gate oxide thickness was the same as the AT-MOSFET, and the thickness and doping concentration of the drift region was set to 12 μm and 5.0 × 10^17^ cm^−3^, respectively. Physical models, including the avalanche generation “Okuto–Crowell Model”, Shockley–Read–Hall (SRH) recombination, temperature and high electric-field-dependent carrier mobility, incomplete ionization, thermodynamics model, and temperature-dependent intrinsic carrier density, are taken into consideration in the simulation.

It is found from Figure 12a that the corner electrical field of the oxide trench in the AT-MOSFET is 2.13 MV/cm, and the maximum I.I is located near to the body diode, while the I.I under the bottom of the oxide trench is below the 10^17^ cm^−3^∙s^−1^, which indicates that deep p-wells in the asymmetric trench structure could effectively limit the electric field crowding at the trench bottom and prevent the gate oxide breakdown [24,25]. The V_DS_ maintains 1650 V during avalanche; the maximum lattice temperature obtained by simulation is 515 K. However, for the DT-MOSFET, the electrical field in Figure 12c under the bottom of the oxide trench is 3.28 MV/cm, the maximum I.I is located around the body diode, and because of a lack of P-type shielding, the I.I can reach up to 10^21^ cm^−3^∙s^−1^. Under the collective effects of the high electrical field, impact ionization, the holes generated by I.I after obtaining a high level of energy, and being accelerated by avalanche, the electrical field would cross the barrier and be injected into the gate oxide, thus becoming hot carriers in the oxide. The avalanche voltage maintains 2050 V. So, for the AT-MOSFET, the reliability of the body diode could be responsible for the failure of the device, while for the DT-MOSFET, both the gate oxide breakdown and lattice temperature are responsible for the failure of the device.

## 4. Optimization and Simulation

In order to reduce the degradations of the SiC MOSFET, especially the DT-MOSFET under repetitive UIS stress, several improved structures have been proposed, as shown in Figure 13. Based on the DT-MOSFET, a new stepped-oxide structure with p-type shielding under the oxide trench and current spreading layer (CSL) beneath the channel is proposed here in Figure 13c. Correspondingly, its electrical field profiles at its avalanche voltage point is shown in Figure 14a–c. Figure 14a gives the electrical field profile of conventional structure, Figure 14b gives electrical field profile of modified DT-MOSFET and Figure 14c gives electrical field profile of the proposed DT-MOSFET. 

Figure 13a depicts a conventional DT-MOSFET. In Figure 13b, a heavy P-type dopant shielding layer is added under the oxide trench to avoid the oxide breakdown in advance. However, P-type shielding will form JFET resistance in conjunction with the P-Well region and lead to an increase in R_on_. In order to resolve this problem, a layer of N-type current spreading is added under the channel. Simulation results have shown that, at avalanche voltage working conditions, the horizontal electrical field beneath the oxide trench could be released; however, at the corner of the oxide trench there is still a high electrical field that exits around 3.12 MV/cm which is shown in Figure 14b. Then, in Figure 13c, a stepped-oxide structure with p-type shielding under the oxide trench and CSL beneath the channel is proposed. The corner of the oxide trench is removed with the substitution of the stepped-oxide corner and is protected by a P-type shielding layer. The electrical field at the corner of the oxide trench is reduced to 2.62 MV/cm which is shown in Figure 14c and benefits from the P-type shielding and collective CSL effect. 

Figure 15 gives the blocking characteristics, V_BV_ and R_on,_ of these three structures. Compared with the conventional DT-MOSFET and modified MOSFET, the R_on_ of the proposed structure decreased about 17.9% with an almost constant breakdown voltage. The breakdown voltage increased from 1320 V to 1460 V, equivalent to about a 10.6% increase. 

Figure 16 presents the suggested feasible processing flow of the proposed stepped-oxide structure with p-type shielding and CSL. Initial P-well/CSL/N-drift/N+substrate is choosen as our substrate. The whole processing flows are totally 12 steps. In real processing flow, extra mask could be added between step (g) and step (h) according to the depth and concentration of P-shield region/P+ region.

## 5. Conclusions

The degradation mechanisms and failure analysis of SiC power MOSFETs under single- and repetitive-frequency UIS characteristics are investigated in this work by experiment and simulation. Previous research has shown that, compared with inductive load, ambient temperature has less influence on the E_AV_ of DUTs. For asymmetric trench devices, avalanche withstanding capability slightly decreases with increasing temperature, while for dual trench devices, avalanche withstanding capability slightly increases with increasing temperature. Under a repetitive UIS test, for an AT-MOSFET with a 20% avalanche energy ratio, V_th_ and R_on_ decreased about 3.4% and 2%, respectively, and C_gd_ increased by 16.7% while the gate oxide remained stable, which indicates that the degradation of devices could be caused by the increase in interface states due to impact ionization. For the DT-MOSFET, V_th_ and R_on_ decreased by about 5.4% and 5.1%, respectively, under the same avalanche energy ratio. The gate leakage current I_gss_ is 7 nA, which will increase with a higher avalanche energy ratio. Thus, the degradation of DT-MOSFETs under repetitive pulse is attributed to the hot carrier that is injected into the oxide, leading to the increase in gate leakage current and damage to the gate oxide.

Furthermore, I.I and the electrical field profile at the avalanche state obtained by simulation have shown that the I.I under the oxide trench of the DT-MOSFET will be higher than that of the AT-MOSFET, which further proves that the gate oxide of the DT-MOSFET could be damaged more by electro-thermal stress than the AT-MOSFET.

Additionally, failed devices were decapsulated to verify the location of the failure point and further confirmed the consistencies between the simulation and the experiment. Finally, a new type of stepped-oxide vertical power DT MOSFET with P-type shielding and current spread layers, along with its feasible process flow, is proposed for the improvement of gate dielectric reliability. Compared with DT-MOSFETs, R_on_ could be decreased by 17.9% with almost the same avalanche voltage ratio. 

## Figures and Tables

**Figure 1 micromachines-15-00772-f001:**
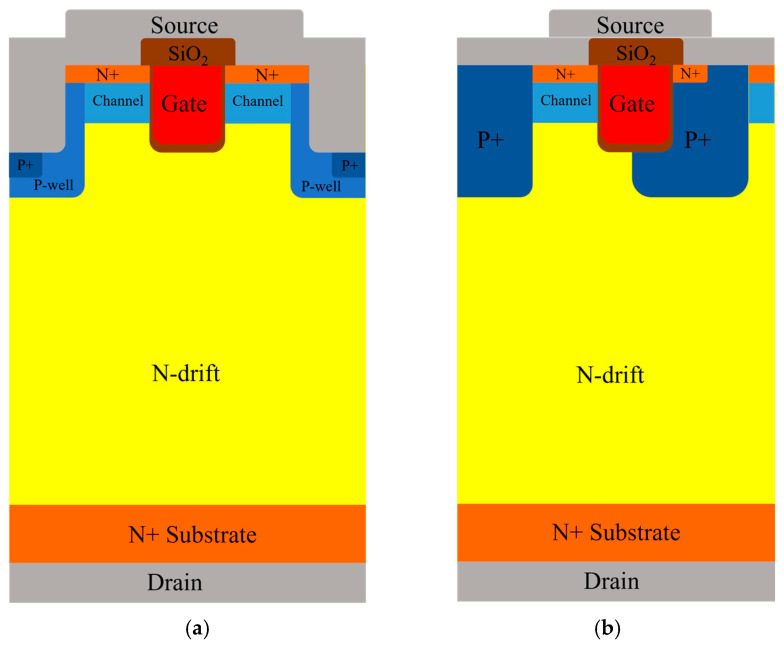
Cross-section of 4H-SiC trench MOSFET: (**a**) DT-MOSFET; (**b**) AT-MOSFET.

**Figure 2 micromachines-15-00772-f002:**
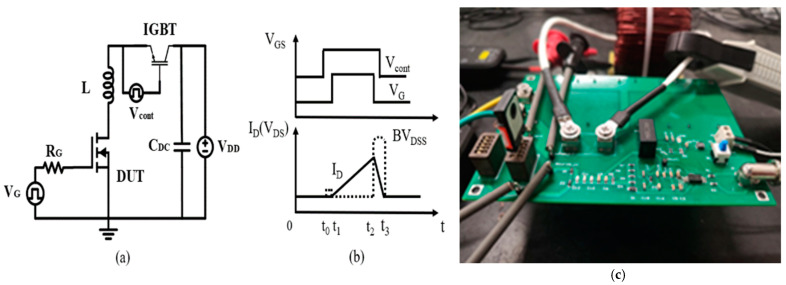
UIS test circuit of 4H-SiC power MOSFET: (**a**) Schematic circuit of UIS test; (**b**) wave form diagram of tested circuit; (**c**) UIS test setup.

**Figure 3 micromachines-15-00772-f003:**
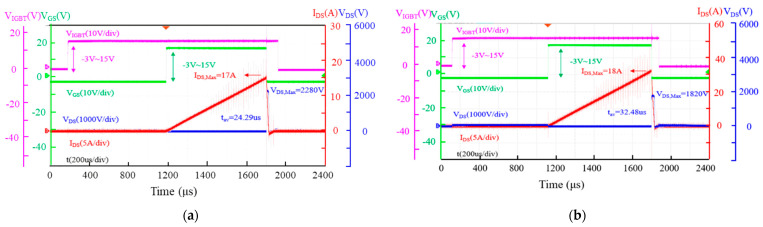
Single-pulse UIS waveforms with load inductor of 3 mH at the last test before failure: (**a**) DT-MOSFET; (**b**) AT-MOSFET.

**Figure 4 micromachines-15-00772-f004:**
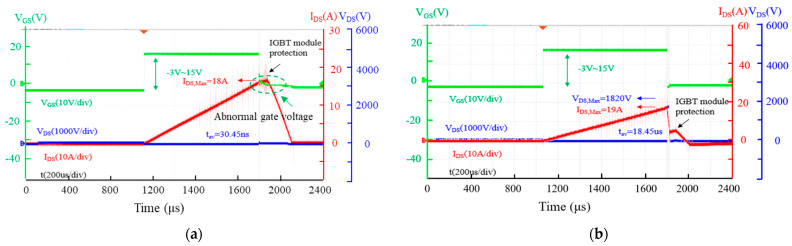
Single-pulse UIS waveforms with load inductor of 3 mH at failure: (**a**) DT-MOSFET; (**b**) AT-MOSFET.

**Figure 5 micromachines-15-00772-f005:**
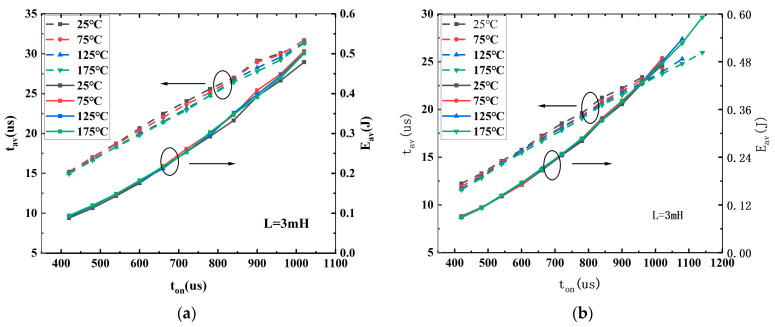
E_AV_ and t_AV_ versus t_on_ for AT-MOSFETs (**a**) and DT-MOSFETs (**b**) under different ambient temperatures with inductor load of 3 mH.

**Figure 6 micromachines-15-00772-f006:**
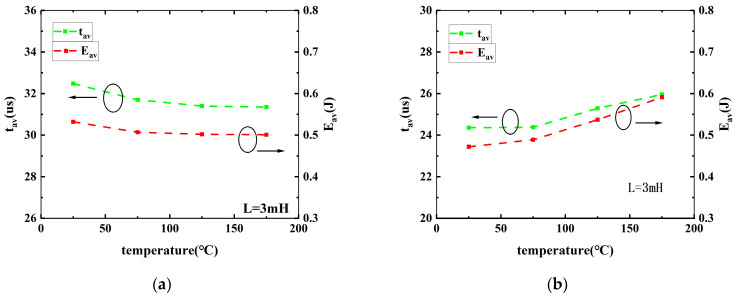
E_AV_ and t_AV_ versus ambient temperature of AT-MOSFETs (**a**) and DT-MOSFETs (**b**).

**Figure 7 micromachines-15-00772-f007:**
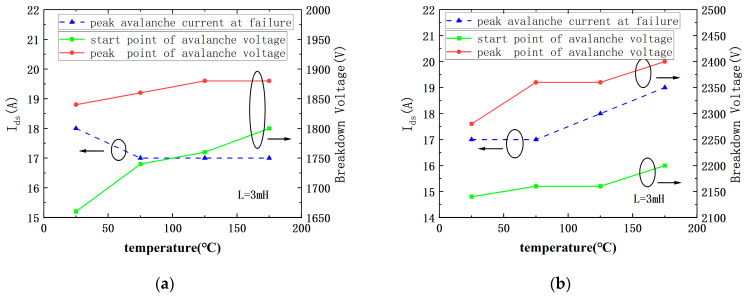
I_peak_, BV_st_, and V_peak_ of DT-MOSFETs (**b**) and AT-MOSFETs (**a**) versus ambient temperature at failure with load inductor of 3 mH.

**Figure 8 micromachines-15-00772-f008:**
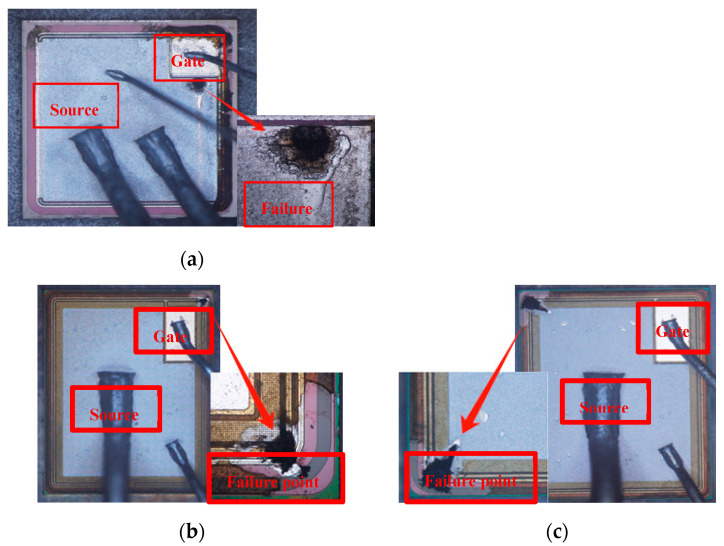
Decapsulation image of AT-MOSFET (**a**), decapsulation of DT-MOS and failure point at 25 °C (**b**), and decapsulation of DT-MOS and failure point at 175 °C (**c**).

**Figure 9 micromachines-15-00772-f009:**
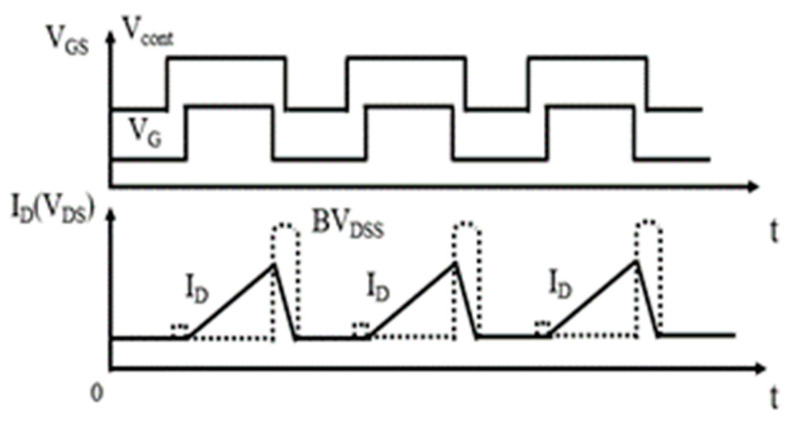
Repetitive avalanche UIS test waveform diagrams.

**Figure 10 micromachines-15-00772-f010:**
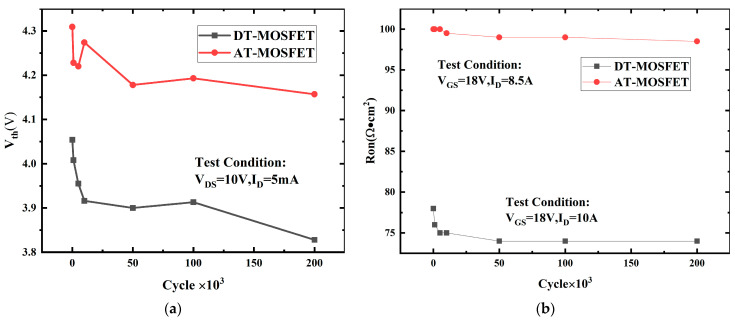
V_th_ versus cycles (**a**) and R_on_ versus cycles (**b**), V_th_ and R_on_ of AT-MOSFET (0.46 × 0.2 = 0.09 J, L = 3 mH) and DT-MOSFET (0.19 × 0.2 = 0.038 J, L = 3 mH) versus cycles under 20% avalanche energy ratio.

**Figure 11 micromachines-15-00772-f011:**
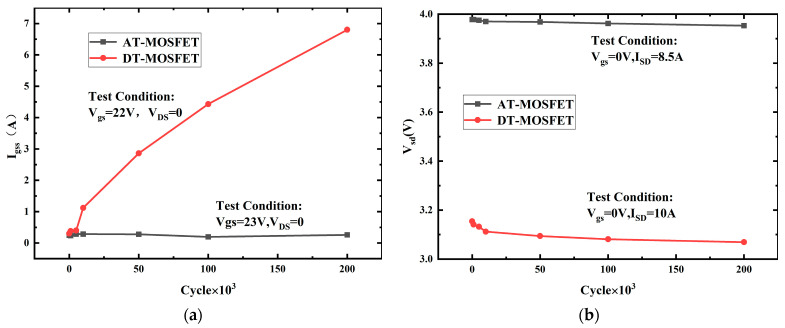
I_gss_ versus cycles (**a**) and V_sd_ versus cycles (**b**), gate leakage current (I_gss_) and forward voltage of body diode (V_sd_) of AT-MOSFET and DT-MOSFET under 20% avalanche energy ratio.

**Figure 12 micromachines-15-00772-f012:**
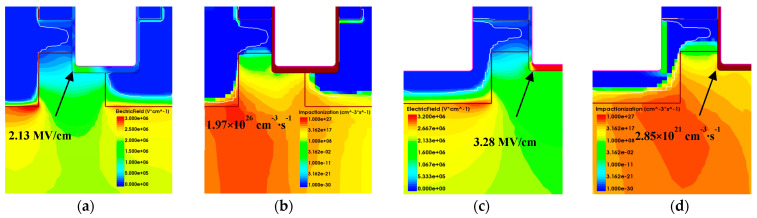
Electrical field of AT-MOSFET (**a**), I.I of AT-MOSFET (**b**), electrical field of DT-MOSFET (**c**), I.I of DT-MOSFET (**d**), and electrical field profile and impaction ionization (I.I) of AT-MOSFET and DT-MOSFET under 20% avalanche energy ratio.

**Figure 13 micromachines-15-00772-f013:**
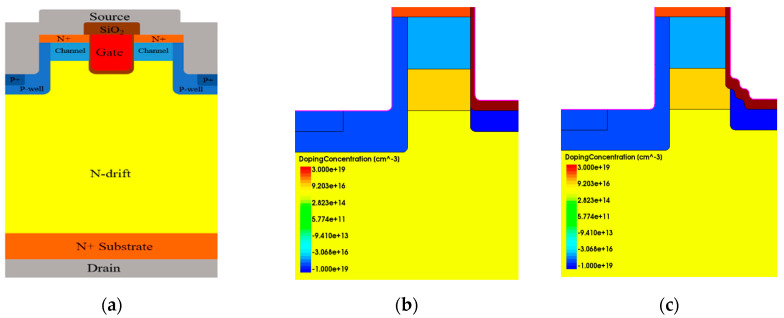
Cross-section comparison of three devices: (**a**) conventional DT-MOSFET, (**b**) modified DT-MOSFET, and (**c**) proposed DT-MOSFET.

**Figure 14 micromachines-15-00772-f014:**
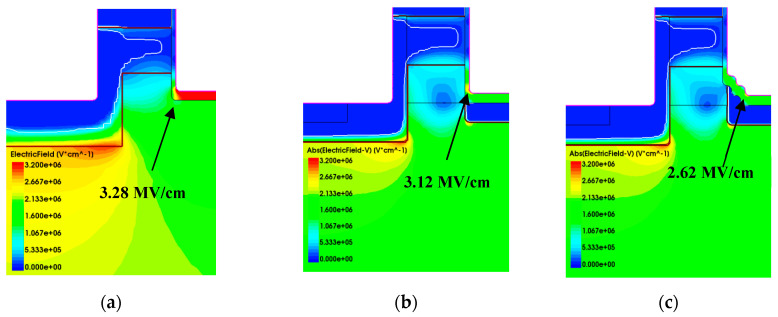
Electrical field profile in different structures (obtained at avalanche points): (**a**) conventional DT-MOSFET, (**b**) modified DT-MOSFET, and (**c**) proposed DT-MOSFET.

**Figure 15 micromachines-15-00772-f015:**
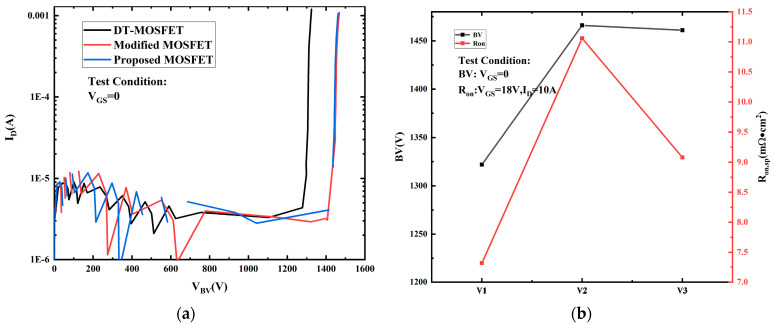
Blocking characteristics and R_on_ variation of these three devices (**a**) Blocking characteristics, (**b**) R_on_ variation of these three devices.

**Figure 16 micromachines-15-00772-f016:**
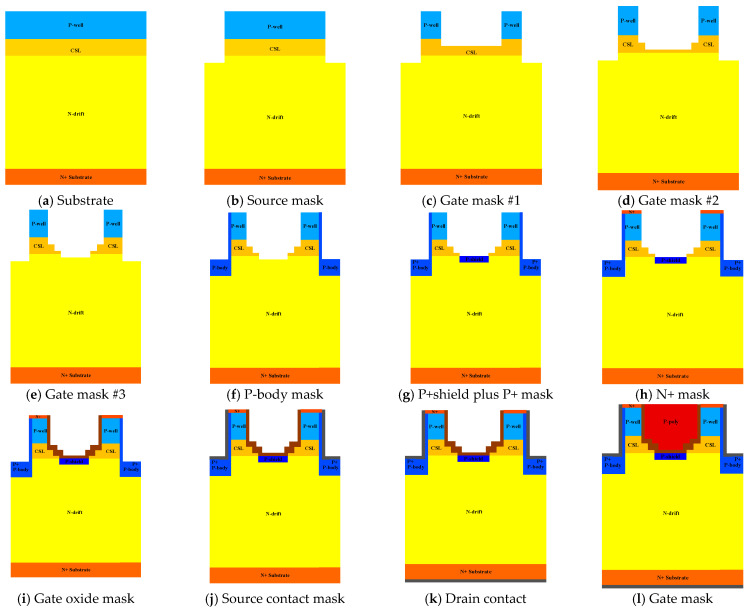
Feasible process flow of proposed DT-MOSFET in Figure 13c.

**Table 1 micromachines-15-00772-t001:** Comparison of E_AV_, I_DS,MAX_, V_DSMAX,_ and t_AV_ tested under single UIS pulse with different inductor loads at room temperature.

Parameters	DT-MOSFET			AT-MOSFET		
	1 mH	3 mH	10 mH	1 mH	3 mH	10 mH
E_AV_	0.19 J	0.47 J	0.57 J	0.46 J	0.53 J	0.67 J
I_DS,max_	19 A	18 A	10.5 A	30 A	19 A	11 A
V_DS,Max_	2240 V	2280 V	2320 V	1840 V	1820 V	1800 V
t_AV_	3.13 μs	30.45 ns	49.35 μs	15.23 μs	18.45 μs	40 μs

## Data Availability

Data is contained within the article.

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
