# Peer review of "A Comprehensive Analysis of Unclamped-Inductive-Switching-Induced Electrical Parameter Degradations and Optimizations for 4H-SiC Trench Metal-Oxide-Semiconductor Field-Effect Transistor Structures"

_micromachines, 2024, doi:10.3390/mi15060772_

Round 1

Reviewer 1 Report

Comments and Suggestions for Authors

This paper presents comprehensive research on Unclamped-Inductive-Switching (UIS)-Induced electrical parameters degradations and optimizations for 4H-SiC trench MOSFETs structures. The work is valuable and interesting. The manuscript could make meaningful contributions to device selection. However, for publication this journal, the authors need to address some issues and improve the explanations of the results.

1. It is suggested to give the structural parameters of the reference device as shown in Fig. 1.

2. Table 1 show tAV tested under single UIS pulse with different inductor load at room temperature. Please explain the abnormal avalanche time tAV at 3 mH about DT-MOSFET.

3. In Fig.12, the same Electrical field scale should be adopted for a better comparison.

4. I’ll praise it if you could show the improved structures feasible process flow diagram.

5. To better calibrate the model, Figure 15(a) suggests to give the leakage current in logarithmic coordinates.

6. The pictures (Fig. 5, 6, 7, 10 and 15) in the text are so blurry. I'll praise if they can be re-illustrated.

Comments on the Quality of English Language

 The paper needs a review from the text (English) point of view. For example: page8, 254.

Reviewer 2 Report

Comments and Suggestions for Authors

This manuscript discusses avalanche failure mechanisms for 4H-SiC MOSFET devices, specifically looking at the effect of temperature and load inductance. The authors have used a combination of experiment utilizing commercially available chips in their own custom circuit as well as numerical simulations on the device structures. Failure is parametrized in this study through the t_AV, avalanche time, and E_AV, avalanche energy, variables. Through these experiments, the authors show little-to-no dependence of both t_AV and E_AV on device temperature, but a quite large dependence on load inductance as well as they increase drastically with the on-state time. Numerical simulations were done to provide supporting information on a mesoscopic scale to look at internal electric field profiles at the point of failure. The authors conclude by proposing new SiC MOSFET designs to reduce degradation in dual-trench MOSFET devices.

The manuscript is well laid out and informative. The study is overall thorough and well put together. I would recommend publication after fixing some minor issues regarding the quality of English grammar in the text and some minor figure edits. Please see below for a list of specific points I found while reading the manuscript.

Lines 25-27 need citation(s)

Paragraph starting at line 28 has significant English grammar issues. Fractured clauses, improper tenses of words, and run-on sentences. I highlighted the issues as best I could in pink.

Lines 49-50 – what are examples of important loop parameters that have been previously studied?

Line 51 – I think you forgot the word ‘study’ after ‘comprehensive’

Figures 3 and 4 should have their axes labeled.

Line 117 – “occure” is spelled “occur”. “Increasement” is not a word, do you mean increase?

Figure 5 is extremely confusing in it’s current state. What do the different symbols represent? The legend is barely legible and the differentiation between which dataset is for t_AV and E_AV is unclear.

Comments on the Quality of English Language

I've included a highlighted PDF of the manuscript with all of the points of poor English grammar or misspelling I noticed. In general, the discussion of results was fine, but the grammar in the introduction was poor.

Round 2

Reviewer 1 Report

Comments and Suggestions for Authors

This revised manuscript can be accepted for publication.